# Enhanced Effect of Patient Room Disinfection Against Carbapenem-Resistant *Enterobacter cloacae* and Methicillin-Resistant *Staphylococcus aureus* Using UV-C Irradiation in Conjunction with UV-C Containment Unit

**DOI:** 10.3390/antibiotics13121115

**Published:** 2024-11-22

**Authors:** Shiori Kitaya, Kentarou Takei, Yoshitomo Honda, Risako Kakuta, Hajime Kanamori

**Affiliations:** 1Department of Infectious Diseases and Laboratory Medicine, Kanazawa University, Kanazawa 920-8641, Japan; 2Department of Otolaryngology, Head and Neck Surgery, Tohoku University Graduate School of Medicine, Sendai 980-8574, Japan; kakuta-r@med.tohoku.ac.jp; 3Department of Infectious Diseases, Internal Medicine, Tohoku University Graduate School of Medicine, Sendai 980-8575, Japan; kentarou.takei.e7@tohoku.ac.jp; 4Moraine Corporation, Nakano 164-0003, Japan; y_honda@moraine.co.jp

**Keywords:** ultraviolet-C, UV-C containment unit, environmental transmission, carbapenem-resistant *Enterobacter cloacae*, methicillin-resistant *Staphylococcus aureus*

## Abstract

**Background/Objectives**: In environments with high-frequency contact surfaces, drug-resistant bacteria, such as carbapenem-resistant *Enterobacterales* and methicillin-resistant *Staphylococcus aureus* (MRSA), can survive for extended periods, contributing to healthcare-associated infections. Ultraviolet (UV)-C irradiation often fails to adequately disinfect shadowed areas, leading to a persistent contamination risk. We evaluated the effectiveness of using a UV-C containment unit (UVCCU) in conjunction with UV-C irradiation to improve the sterilization effects on both direct and indirect surfaces, including shadowed areas, and to assess the leakage of UV radiation to the surroundings. **Methods**: In a model patient room, agar media inoculated with carbapenem-resistant *Enterobacter cloacae* and MRSA were placed at multiple locations on direct and indirect surfaces around the bed. We used the UV-C irradiation system, UVDI-360, to irradiate the bedroom-environment surfaces with and without a UVCCU. The reduction in bacterial colony counts with and without the UVCCU was measured by counting colony-forming units and calculating the log reduction values, and the UV radiation leakage outside the UVCCU was measured. **Results**: The use of the UVCCU led to a significant reduction in MRSA colony counts, even in shadowed areas that had previously been inadequately disinfected (with the UVCCU: 2.7 [2.7–2.8]; without the UVCCU: 0.6 [0.5–0.7]; *p* < 0.01). Additionally, the use of the UVCCU kept the UV radiation leakage to the surrounding environment within regulated limits. **Conclusions**: These findings suggest that a UVCCU can enhance the disinfection efficacy for multidrug-resistant organisms on healthcare environmental surfaces. The portability and ease of use of the UVCCU indicate its promise as an auxiliary device for UV-C disinfection in healthcare settings.

## 1. Introduction

In environments, particularly those with high-frequency contact surfaces, drug-resistant bacteria can survive for extended periods. The transmission of drug-resistant bacteria can contribute to the occurrence of healthcare-associated infections (HAIs) [1]. Over 25% of HAIs are caused by cross-contamination via contaminated environmental surfaces [2], and outbreaks of multidrug-resistant bacterial infections from environmental surfaces have been reported worldwide [3,4]. Additionally, when a new patient is admitted to a room previously occupied by a patient infected or colonized with multidrug-resistant bacteria, the risk of acquiring methicillin-resistant *Staphylococcus aureus* (MRSA), vancomycin-resistant *Enterococci*, multidrug-resistant *Acinetobacter*, and other drug-resistant bacteria increases by 39–353% [5,6].

High-frequency contact surfaces within patient rooms, such as bed rails and tray tables, are the highest-risk areas for HAIs, and appropriate disinfection of these surfaces can prevent HAIs [7]. One of the most common drug-resistant bacteria that can cause HAIs is carbapenem-resistant *Enterobacterales* (CRE) [8,9]. CRE infections have limited effective antibiotic options and, when they lead to bloodstream or severe infections, result in poor prognosis and high mortality rates [8,10]. Hospital CRE outbreaks have also been reported globally [11].

Recently, “no-touch” environmental disinfection methods, such as ultraviolet (UV) light and hydrogen peroxide vapor systems, have been introduced as adjuncts to traditional manual cleaning [12,13]. Incorporating UV-C devices into standard cleaning and disinfection procedures can further reduce the microbial load on contaminated surfaces and improve disinfection levels in healthcare settings [14,15]. UV-C devices have shown efficacy against CRE, with direct UV exposure reducing carbapenem-resistant *Klebsiella pneumoniae* counts by over 99.999%, and indirect exposure reducing counts by over 99.99% [16].

The effectiveness of UV-C disinfection technologies depends significantly on environmental factors, such as the distance from the radiation source, UV-C intensity, and arrangement of objects in the room. UV-C may be less effective in shadowed areas to which light does not directly reach [17]. The significant side effects of UV-C exposure include sunburn-like reactions and photokeratitis; therefore, it is necessary for both users and patients to stay clear of the irradiation area when using UV-C devices [18]. For example, a case of acute sunburn caused by an electric insect killer using UV-C, with an irradiance of up to 46 mW m^−2^, has been reported [19]. When multiple patients share the same room, it is challenging to irradiate a section of the room when one patient is discharged, particularly if other patients are severely ill or have difficulty moving.

To overcome the difficulties of irradiating a specific section of a shared hospital room, a UV-C containment unit (UVCCU) was developed. A UVCCU is designed to shield against UV rays during irradiation, allowing for partial and efficient UV irradiation of shadowed areas within the room while also preventing UV leakage outside the UVCCU. This study aimed to evaluate (i) the log reduction in colony numbers of carbapenem-resistant *Enterobacter cloacae* and MRSA due to UV irradiation using the UV-C irradiation system, UVDI-360, with and without a UVCCU and (ii) UV leakage outside the shielded area of the UVCCU.

## 2. Results

### 2.1. Evaluation of Colony Reduction Rate by Log Reduction

The colony reduction rates by log reduction for carbapenem-resistant *E. cloacae* and MRSA are shown in Table 1. In the bed rail area for carbapenem-resistant *E. cloacae*, one of the three irradiation cycles showed clear contamination and was excluded from the analysis, leaving two irradiation cycles included in the analysis. The log reduction for carbapenem-resistant *E. cloacae* on five direct surfaces and two indirect surfaces (three irradiation cycles, n = 20) was 2.3 (2.2–2.3) with the UVCCU and 2.3 (2.1–2.4) without the UVCCU, showing no significant difference. The log reduction on the direct and indirect surfaces also showed no significant differences with or without the UVCCU.

For MRSA, the log reduction on five direct surfaces and two indirect surfaces (three irradiation cycles, n = 21) was 2.7 (2.7–2.8) with the UVCCU and 2.0 (1.6–2.5) without the UVCCU. The presence of the UVCCU significantly reduced MRSA growth across the irradiated surfaces compared with that in the absence of the UVCCU (*p* < 0.01). On indirect surfaces (three irradiation cycles, n = 6), MRSA growth was also significantly reduced with the UVCCU compared to that without the UVCCU (with the UVCCU: 2.7 [2.7–2.8]; without the UVCCU: 0.6 [0.5–0.7]; *p* < 0.01). Additionally, among the direct surfaces on the overbed table (tabletop) (three irradiation cycles, n = 3), MRSA growth was significantly reduced with the UVCCU compared to that without the UVCCU (with the UVCCU: 2.8 [2.8–2.8]; without the UVCCU: 2.3 [1.4–3.3]; *p* = 0.04).

### 2.2. The UV Radiation Levels Around the UVCCU

The UV radiation levels around the UVCCU, measured using a UV intensity meter, are listed in Table 2. The UV radiation levels around the UVCCU were approximately 99.9% lower with the UVCCU compared to that without the UVCCU. In addition, the UV radiation levels around the UVCCU were within the standards specified by JIS Z8812, a Japanese Industrial Standard that establishes safety guidelines for UV radiation.

## 3. Discussion

### 3.1. Change in Colony Counts Before and After Irradiation with and Without UVCCU

Various pathogens adhere to surfaces in the medical environment and can transfer to the hands of healthcare workers and patients through contact with these surfaces [1]. Therefore, infection control in healthcare facilities places great importance on cleaning environmental surfaces in addition to hand hygiene. However, manual cleaning with chemical agents can vary significantly in terms of methods and effectiveness, with reports indicating that up to 50% of high-touch surfaces in patient areas may be missed because of restricted access or human error during chemical cleaning [20]. Recently, new disinfection methods using devices such as vaporized hydrogen peroxide generators and automated UV irradiation devices have gained attention as adjuncts to traditional manual cleaning methods. In particular, UV-C devices do not require changes to room ventilation, leave no residue after treatment, have a broad action spectrum, and provide rapid exposure times [17,21].

The UV-C irradiation system used in this study, the UVDI-360, is equipped with four high-output UV lamps, each 1580 mm in length. It features a simple programming and operation interface, is mobile for easy relocation, and includes four infrared motion sensors that halt operation immediately upon detection of human intrusion, making it highly useful [22]. However, the effectiveness of UV-C disinfection technology has been reported to depend significantly on environmental factors, such as the distance from the radiation source, UV-C intensity, and the arrangement of objects in the room. This may be insufficient in shadowed areas to which light does not directly reach [17]. In this study, combining the UVCCU with UV irradiation using UVDI-360 achieved a significant disinfection effect against MRSA, even in shadowed areas previously considered difficult for UV light to reach. This is presumably due to the reflective effect of the UVCCU. The proactive use of the UVCCU is expected to improve the disinfection effectiveness of environmental surfaces around patients, including previously inadequately disinfected shadowed areas, and ultimately reduce the frequency of HAIs caused by pathogen transmission from environmental surfaces.

### 3.2. The Effectiveness of UVCCU in Suppressing UV Leakage to the Surroundings 

UV-C radiation has high energy and can cause severe sunburn-like reactions or UV keratitis upon direct exposure [18]. Therefore, it is necessary to leave the room when using UV-C light devices. However, when a patient with drug-resistant bacteria is discharged from a multibed hospital room, it can be challenging to quickly irradiate only that section, especially if the other patients are seriously ill or have difficulty moving. In this study, we found that UV-C leakage outside the UVCCU was within regulatory standards when UV-C irradiation was performed using the UVCCU. Therefore, using a UVCCU could allow partial and efficient UV-C irradiation of a specific section of a multibed hospital room without requiring other patients to be moved. Additionally, a UVCCU can prevent UV leakage into the surroundings, allowing irradiation with sufficient UV intensity for disinfection without affecting the surrounding environment. A UVCCU can be compactly folded, is lightweight, and easy to store and transport within a hospital. In the future, we expect that UVCCUs will be widely used as essential auxiliary devices for UV-C in healthcare facilities.

### 3.3. Limitations

This study had several limitations, such as the small sample size and limited number of bacterial species examined. However, it is considered an important study as it demonstrated the enhanced bactericidal effects when using the UVCCU in conjunction with UVDI-360 irradiation and the UV shielding effectiveness of the UVCCU.

## 4. Materials and Methods

### 4.1. Setting the Irradiation Environment

A model patient room (4-bed room) was created at Tohoku University Hospital. A bed unit that mimicked an actual hospital room was set up as one of the beds, and a shielding system, the UVCCU, was installed around the bed unit (Figure 1). The UV-C irradiation system used was UVDI-360 (Moraine Corporation, Nakano, Tokyo, Japan), which has a wavelength of 254 nm and is equipped with four 1580 mm lamps. The UVDI-360 is characterized by its extremely simple program settings and operation, mobility, ease of movement, and safety features; it is equipped with four infrared motion sensors that immediately stop the system when human presence is detected [23]. The UVCCU was designed as a shield against UV irradiation and created to enable the use of UV-C irradiation devices for terminal cleaning at the time of patient discharge in multibed rooms by temporarily shielding the area of the discharged patient, allowing other patients in the same room to remain present. By shielding the area around a single bed with the unit in a multibed room, it is possible to irradiate areas that are difficult to expose directly to UV light by utilizing the reflection effect within the unit. In addition, the unit prevents UV light from leaking outside the shielded area. When fully deployed, its dimensions are approximately 2700 mm × 3000 mm, and it weighs approximately 312 g. It is compactly folded for easy transport and the set up time is approximately 15 min. It can accommodate UV irradiation up to a height of 2.5 m and can be used to sterilize various medical instruments. While the standard configuration provides four-sided shielding, it can be adjusted to two- or three-sided shielding, depending on the conditions of the surrounding area.

Agar media inoculated with carbapenem-resistant *E. cloacae* and MRSA were placed at seven locations, including direct and indirect surfaces exposed to UV-C. The following locations are indicated in Figure 2 as direct surfaces: (1) head position of the bed, (2) overbed table (tabletop), (3) bedside cabinet (tabletop), (5) bed rails, and (7) curtain (pole part). The following are indicated as indirect surfaces: (4) footboard (outer side) and (6) bedside cabinet (left side).

### 4.2. Preparation of Bacterial Suspension

A culture medium inoculated with the target bacterial species (clinical isolates of carbapenem-resistant *E. cloacae* and MRSA detected based on culture tests in patients who visited Tohoku University) was used as an environmental substitute carrier and placed in seven locations where UV-C was irradiated according to the specified method. After the completion of the UV-C cycle, these plates were incubated aerobically at 37 °C for 48 h, and the colony-forming units (CFU) of carbapenem-resistant *E. cloacae* and MRSA were objectively counted. Each bacterium was cultured on a standard agar medium at 37 °C for 24 h. On the day of the experiment, a single colony of each bacterium was selected and inoculated into 0.85% sterile saline to prepare a bacterial suspension with an OD600 of 0.5. One milliliter of this suspension was inoculated into 9 mL of sterile saline to create a primary dilution, followed by subsequent dilutions up to the fifth dilution. Then, 0.1 mL of the fifth dilution was applied to the standard agar medium. Media preparation, sample adjustment, and colony counting were performed by Japan Biosciences Co., Ltd. (Osaka, Japan).

### 4.3. UV Irradiation and Evaluation Methods

The UVDI-360 was placed at two locations designated as points A and B (Figure 3). One cycle consisted of irradiation at point A for 5 min, followed by 5 min at point B. This was repeated three times with and without the UVCCU. The medium was collected after each UV-C irradiation cycle and incubated for a specified duration. Subsequently, the CFU of carbapenem-resistant *E. cloacae* and MRSA were counted, and the reduction in colony numbers before and after irradiation with and without the UVCCU was evaluated using log reduction values [16]. The log reduction was calculated by taking the logarithm of the ratio of the colony count of the control to that after irradiation. The control was a standard agar medium inoculated with a diluted bacterial suspension without UV irradiation.

### 4.4. Measurement of UV Radiation Outside UVCCU

The UV intensity outside the UVCCU was measured seven times using ILT2400 and SED240/QNDS2/TD (International Light Technologies, Peabody, MA, USA). The UV meter was positioned at (X: −100 cm, Y: −125 cm, Z: 165 cm) from point O, with the receiving surface oriented toward the direction where the ceiling and the shielding curtain overlap. The UV radiation levels around the UVCCU were based on the allowable range for 254 nm UV radiation (≤6 mJ/cm^2^), as specified by the safety guidelines in the Japanese Industrial Standard (JIS Z8812) [24].

### 4.5. Evaluation Criteria

The primary endpoint was the log reduction in the colony numbers of carbapenem-resistant *E. cloacae* and MRSA due to UV irradiation using the UVDI-360 with and without UVCCU shielding. The secondary endpoint was the assessment of UV leakage outside the shielded area of the UVCCU.

### 4.6. Statistical Analysis

Colony reduction was calculated by taking the logarithm of the ratio of colony count in the control (numerator) to colony count after irradiation (denominator). The Wilcoxon rank-sum test was used to compare the log reduction values of carbapenem-resistant *E. cloacae* and MRSA caused by UV irradiation with the UVDI-360 between the groups with and without the UVCCU. The analysis was performed using the JMP Pro 17 statistical analysis software (SAS Institute, Cary, NC, USA). Differences were considered significant if the *p*–value was <0.05.

## 5. Conclusions

In this study, the use of the UVCCU in conjunction with UV-C irradiation resulted in a significant bactericidal effect on MRSA, even in areas that were previously considered shadowed and difficult for UV light to reach. Furthermore, the use of the UVCCU kept the UV radiation leakage to the surrounding environment within regulated limits.

By actively utilizing a UVCCU in the future, it is expected that the disinfection effectiveness of surfaces in the medical environment around patients, including shadowed areas, will be enhanced, thereby reducing the frequency of HAIs caused by pathogen transmission from environmental surfaces. Moreover, the use of a UVCCU could enable partial and efficient UV-C irradiation of a specific section in a multibed hospital room without requiring other patients to be moved.

## Figures and Tables

**Figure 1 antibiotics-13-01115-f001:**
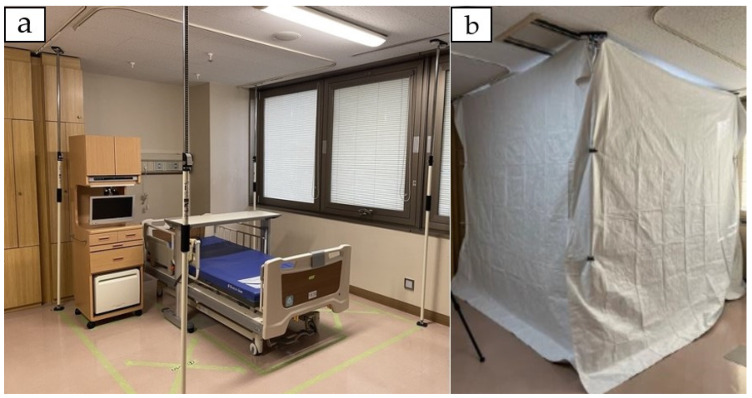
The bed unit and shielding system UVCCU. (**a**) A bed unit that mimics an actual hospital room is created in one bed of the model hospital room (4-bed room); (**b**) a shielding system UV-C containment unit is installed to surround the bed unit. UV, ultraviolet; UVCCU, UV-C containment unit.

**Figure 2 antibiotics-13-01115-f002:**
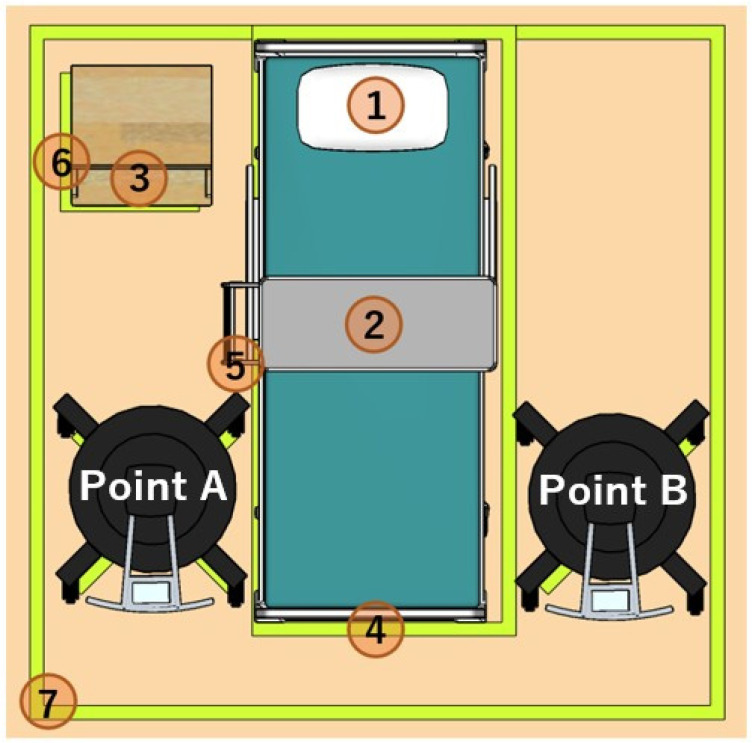
The locations for placing agar media and the position of the UVDI-360 within the bed unit. Agar media coated with carbapenem-resistant *Enterobacter cloacae* and methicillin-resistant *Staphylococcus aureus* were placed at seven locations, including surfaces directly hit by UV-C and indirect surfaces that were shaded. The distances from the UVDI-360 installed on the right and left sides of the bed to each directly irradiated surface were as follows: head position of the bed, right (171 cm), left (173 cm); overbed table (tabletop), right (105 cm), left (107 cm); bedside cabinet (tabletop), right (123 cm), left (216 cm); bed rails, right (61 cm), left (142 cm); curtain (pole part), right (87 cm), left (226 cm). On the other hand, the distances from the UVDI-360 installed on the right and left sides of the bed to each indirectly irradiated surface were as follows: footboard (outer side), right 92 cm, left 94 cm; bedside cabinet (left side), right (126 cm), left (232 cm). The UVDI-360 was irradiated at position A for 5 min and then at position B for 5 min, considering this as one cycle, and a total of three cycles were conducted. UV, ultraviolet.

**Figure 3 antibiotics-13-01115-f003:**
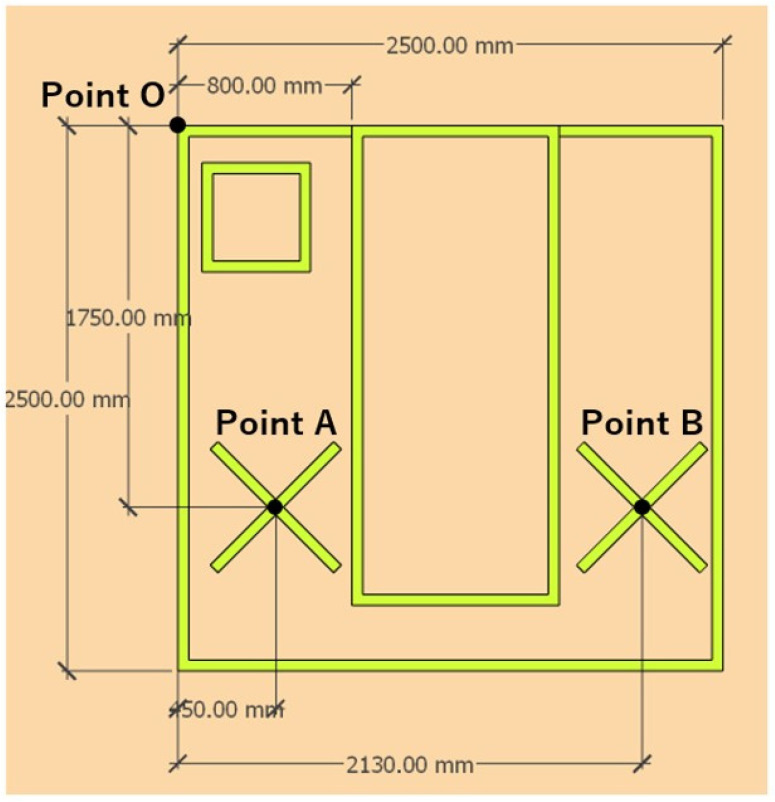
The positional relationship and irradiation method of the UVDI-360 within the bed unit. The UVDI-360 was installed at two locations, designated as points A and B. The UVDI-360 was irradiated at position A for 5 min and then at position B for 5 min, considering this as one cycle, and a total of three cycles were conducted.

**Table 1 antibiotics-13-01115-t001:** Microbial log reduction on surfaces contaminated with MRSA or Carbapenem-resistant *E. cloacae* under UV-C exposure.

UV–C Irradiation Surface	With UVCCU	Without UVCCU	*p*-Value
**Carbapenem-resistant *Enterobacter cloacae***			
Direct Surface (n = 14)	2.2 (2.1–2.4)	2.3 (2.1–2.4)	
head position of the bed (n = 3)	2.3 (2.3–2.3)	2.4 (2.4–2.4)	
overbed table (tabletop) (n = 3)	2.3 (2.3–2.3)	2.3 (2.3–2.3)	
bedside cabinet (tabletop) (n = 3)	2.3 (2.3–2.3)	2.3 (2.3–2.3)	
bed rails (n = 2)	2.3 (2.3–2.3)	2.3 (2.3–2.3)	
curtain (pole part) (n = 3)	2.0 (0.7–3.3)	2.3 (2.3–2.3)	
Indirect Surface (n = 6)	2.3 (2.3–2.3)	2.1 (1.7–2.5)	
footboard (outer side) (n = 3)	2.3 (2.3–2.3)	2.4 (2.4–2.4)	
bedside cabinet (left side) (n = 3)	2.3 (2.3–2.3)	1.8 (0.8–2.8)	0.04
Total (n = 20)	2.3 (2.2–2.3)	2.3 (2.1–2.4)	
**Methicillin-resistant *Staphylococcus aureus***			
Direct Surface (n = 15)	2.7 (2.7–2.8)	2.6 (2.5–2.8)	
head position of the bed (n = 3)	2.6 (1.6–3.6)	2.6 (2.0–3.3)	
overbed table (tabletop) (n = 3)	2.8 (2.8–2.8)	2.3 (1.4–3.3)	0.04
bedside cabinet (tabletop) (n = 3)	2.8 (2.8–2.8)	2.8 (2.8–2.8)	
bed rails (n = 3)	2.8 (2.8–2.8)	2.5 (1.9–3.2)	
curtain (pole part) (n = 3)	2.7 (2.7–2.7)	2.8 (2.8–2.8)	
Indirect Surface (n = 6)	2.7 (2.7–2.8)	0.6 (0.5–0.7)	<0.01
footboard (outer side) (n = 3)	2.8 (2.8–2.8)	0.7 (0.5–0.8)	0.04
bedside cabinet (left side) (n = 3)	2.7 (2.7–2.7)	0.6 (0.4–0.7)	0.04
Total (n = 21)	2.7 (2.7–2.8)	2.0 (1.6–2.5)	<0.01

The distances from the UVDI-360 installed on the right and left sides of the bed to each directly irradiated surface were as follows: head position of the bed, right (171 cm), left (173 cm); overbed table (tabletop), right (105 cm), left (107 cm); bedside cabinet (tabletop), right (123 cm), left (216 cm); bed rails, right (61 cm), left (142 cm); curtain (pole part), right (87 cm), left (226 cm). On the other hand, the distances from the UVDI-360 installed on the right and left sides of the bed to each indirectly irradiated surface were as follows: footboard (outer side), right 92 cm, left 94 cm; bedside cabinet (left side), right (126 cm), left (232 cm). The values in the table represent log reduction and 95% confidence intervals. Log reduction is calculated by taking the ratio of the colony count in the control (numerator) to the colony count after irradiation (denominator) and then taking the logarithm of that value. *p*-values are listed only for values that showed significant differences. UV, ultraviolet; UVCCU, UV-C containment unit.

**Table 2 antibiotics-13-01115-t002:** Comparison of UV–C radiation levels in surrounding area with and without UVCCU.

Measurement	With UVCCU		Without UVCCU	
	UV Radiation with UVCCU (µJ/cm²)	Time to Reach Limit with UVCCU (min)	UV Radiation Without UVCCU (µJ/cm²)	Time to Reach Limit Without UVCCU (min)
1st	205.9	672	140,600	0.47
2nd	148.7	930	163,900	0.40
3rd	159.5	864	166,300	0.40
4th	175.1	786	184,700	0.36
5th	145.9	948	172,000	0.38
6th	173.6	792	164,900	0.40
7th	151.1	912	187,200	0.35
Average	165.7	834	168,514.3	0.39

The upper limit reach time with the UVCCU is calculated as (6000 μJ ÷ UV dose per irradiation) × cycle time (11 min of UV irradiation + 8 min for UVCCU assembly + 4 min for UVCCU removal). The upper limit reach time without the UVCCU is calculated as (6000 μJ ÷ UV dose per irradiation) × cycle time (11 min of UV irradiation). UV, ultraviolet; UVCCU, UV-C containment unit.

## Data Availability

The datasets used and/or analyzed in this study are available from the corresponding author upon reasonable request.

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
