# Peer review of "Enhanced Effect of Patient Room Disinfection Against Carbapenem-Resistant Enterobacter cloacae and Methicillin-Resistant Staphylococcus aureus Using UV-C Irradiation in Conjunction with UV-C Containment Unit"

_antibiotics, 2024, doi:10.3390/antibiotics13121115_

Round 1

Reviewer 1 Report

Comments and Suggestions for Authors

This study looked at how well a UV-C containment unit (UVCCU) cleans surfaces, even in hard-to-reach places. The results showed that using the UVCCU greatly reduced the number of MRSA germs, even in areas that were not properly cleaned before, while also making sure that UV light did not leak outside safe limits. The article is written well. Language is acceptable. There are some minor points should be considered:

-            Abstract: Please provide core data.

-            Keywords: Please don’t use the same words that you already used for title.

-            Please provide more detail about statistical analysis.

-            The reference list should be extended.

Author Response

This study looked at how well a UV-C containment unit (UVCCU) cleans surfaces, even in hard-to-reach places. The results showed that using the UVCCU greatly reduced the number of MRSA germs, even in areas that were not properly cleaned before, while also making sure that UV light did not leak outside safe limits. The article is written well. Language is acceptable. There are some minor points should be considered:

Abstract: Please provide core data.

Response: We are grateful for your insightful comments. As you kindly pointed out, we have added the colony reduction rate by log reduction in MRSA on indirect surfaces, which is a core result in this study, to the abstract.

“The use of the UVCCU led to a significant reduction in MRSA colony counts, even in shadowed areas that had previously been inadequately disinfected (with the UVCCU: 2.7 [2.7–2.8]; without the UVCCU: 0.6 [0.5–0.7]; p < 0.01). ” (Please see Page 1, Lines 27-30)

Keywords: Please don’t use the same words that you already used for title.

Response: Thank you very much for your valuable suggestion. Indeed, using words from the title as keywords may have drawbacks, such as potentially limiting access by various related fields. However, ultraviolet-C and UV-C containment unit are fundamental elements closely related to the rationale of this study. Moreover, carbapenem-resistant Enterobacter cloacae and methicillin-resistant Staphylococcus aureus are key terms that denote the targeted bacterial species. Therefore, from the perspective of making it easier for readers to understand the focus of this research through the keywords, as well as emphasizing the significance of this paper, we would like to intentionally retain these terms in both the title and keywords. We appreciate your understanding.

Please provide more detail about statistical analysis.

Response: We thank you for this comment. In the Statistical Analysis section, we initially only described the comparison method (Wilcoxon rank-sum test) for log reduction values of UV irradiation with the UVDI-360 for carbapenem-resistant E. cloacae and MRSA. However, to enhance clarity for readers, we have also added an explanation about the log reduction values used to evaluate colony reduction.

“Colony reduction was calculated by taking the logarithm of the ratio of colony count in the control (numerator) to colony count after irradiation (denominator).” (Please see Page 8, Lines 267-268)

The reference list should be extended.

Response: We thank you for this comment. To help readers deepen their understanding of UV-C irradiation, we have added several references, focusing on recent studies.

Reviewer 2 Report

Comments and Suggestions for Authors

The purpose of the research is interesting.  It is not clear after UV which changes happened in bacterial cell.  But it will be great if the authors prove that  on which mechanism the UV-C Irradiation in Conjunction With a UV-C Containment unit affect on the resistant bacteria, for example, by genome sequencing method. The statistical data was illustrated in the tables, it will be more interesting if it will be shown with bar chart.

Author Response

The purpose of the research is interesting.  It is not clear after UV which changes happened in bacterial cell.  But it will be great if the authors prove that on which mechanism the UV-C Irradiation in Conjunction With a UV-C Containment unit affect on the resistant bacteria, for example, by genome sequencing method. The statistical data was illustrated in the tables, it will be more interesting if it will be shown with bar chart.

Response: We thank you for this comment. As you excellently pointed out, considering the mechanisms by which the combination of UV-C irradiation with a UV-C containment unit impacts drug-resistant bacteria is fascinating. Using this study as a foundation, we hope to elucidate these mechanisms through future research. Additionally, we attempted to create a bar graph of the results in Table 1. However, we found that including multiple sets of results made the figure cluttered and difficult to read. In addition, the differences between most values were minimal that they were not easily distinguishable on a bar graph. To make it easier for readers to understand, we would prefer to present the data in its current format.

Reviewer 3 Report

Comments and Suggestions for Authors

Following are some questions and suggestions that need to be discussed in the manuscript by the author before acceptance;

1.      What kind of an effect does the number of irradiation cycles have on the overall effectiveness of UV-C therapy, taking into consideration that one of the irradiation cycles for carbapenem-resistant E. cloacae revealed evident contamination and was therefore removed from the analysis? A considerable improvement in log reduction rates across a variety of surfaces might be achieved by increasing the number of cycles.

2.      During the process of treating carbapenem-resistant E. cloacae and MRSA with UV-C, how does the material of the surfaces (such as plastic, metal, or fabric) influence the log decrease rates of these bacteria? Are there any possibilities that future research may incorporate a broader range of surface materials in order to determine which ones are the most resistant to UV-C disinfection?

3.      The findings of the research suggest that the UVCCU enhances the efficiency of disinfection in places that are shrouded in darkness. How does the effectiveness of UV-C irradiation change depending on the design of the UVCCU, specifically whether it is composed of two sides or four sides? What kind of insights may be gained from a research that compares these different arrangements about how to maximize UV exposure?

4.       What kind of long-term effects does irradiating surfaces with UV-C have on those surfaces that have been subjected to UV disinfection in the past? The risk for recontamination may be better understood if following tests were to examine the permanence of bacterial decrease over time.

5.      The carbapenem-resistant E. cloacae and MRSA were the primary foci of this investigation; however, the UVCCU was also tested against other antibiotic-resistant organisms, such as VRE and multidrug-resistant Acinetobacter. A more complete picture of the UVCCU's efficacy may be obtained by increasing the number of pathogens that are screened.

6.      Line 209-210 (clinical isolates of carbapenem-resistant E. cloacae and MRSA)>>>>>>> The author should describe more details of the source of these pathogens.

7.      Are all figures the standard model dimensions and location-wise?

Author Response

Following are some questions and suggestions that need to be discussed in the manuscript by the author before acceptance;

  1. What kind of an effect does the number of irradiation cycles have on the overall effectiveness of UV-C therapy, taking into consideration that one of the irradiation cycles for carbapenem-resistant E. cloacae revealed evident contamination and was therefore removed from the analysis? A considerable improvement in log reduction rates across a variety of surfaces might be achieved by increasing the number of cycles.

Response: We thank you for this suggestion. As you kindly pointed out, increasing the number of irradiation cycles may improve the log reduction rate of drug-resistant bacteria on environmental surfaces. Although the primary objective of this study was to evaluate the additive effect of the UV-C containment unit on UV-C irradiation, we would like to explore, in future studies, how an increased number of irradiation cycles could further impact the log reduction rate of drug-resistant bacteria.

  1. During the process of treating carbapenem-resistant E. cloacae and MRSA with UV-C, how does the material of the surfaces (such as plastic, metal, or fabric) influence the log decrease rates of these bacteria? Are there any possibilities that future research may incorporate a broader range of surface materials in order to determine which ones are the most resistant to UV-C disinfection?

Response: We appreciate your concerns about this point. As you pointed out, the impact of different surface materials, such as plastic, metal, and fabric, on UV-C irradiation effectiveness is indeed very intriguing. In future studies, we would like to investigate the effects of UV-C irradiation under consistent irradiation conditions, varying only the material type, to further understand how different surfaces influence its efficacy.

  1. The findings of the research suggest that the UVCCU enhances the efficiency of disinfection in places that are shrouded in darkness. How does the effectiveness of UV-C irradiation change depending on the design of the UVCCU, specifically whether it is composed of two sides or four sides? What kind of insights may be gained from a research that compares these different arrangements about how to maximize UV exposure?

Response: Thank you for your valuable query. Although this study has confirmed the effectiveness of UV-C containment unit in dark environments, understanding how the effectiveness of UV-C irradiation varies between a two-sided and four-sided UV-C containment unit configuration is also an intriguing question. Moving forward, we would like to conduct comparative studies under different panel conditions to determine the optimal design for maximizing UV exposure. If this research clarifies the panel configuration that maximizes the log reduction rate of drug-resistant bacteria, it could contribute to a more effective utilization of UVCCU.

  1. What kind of long-term effects does irradiating surfaces with UV-C have on those surfaces that have been subjected to UV disinfection in the past? The risk for recontamination may be better understood if following tests were to examine the permanence of bacterial decrease over time.

Response: We thank you for this comment. As you noted, understanding the long-term effects of UV-C irradiation is crucial for determining the optimal frequency for UV-C applications. Moving forward, we aim to conduct ongoing testing to evaluate the persistence of bacterial reduction over time, which will help demonstrate the risk of recontamination and preventive efficacy of UV-C irradiation.

  1. The carbapenem-resistant E. cloacae and MRSA were the primary foci of this investigation; however, the UVCCU was also tested against other antibiotic-resistant organisms, such as VRE and multidrug-resistant Acinetobacter. A more complete picture of the UVCCU's efficacy may be obtained by increasing the number of pathogens that are screened.

Response: Thank you for your valuable feedback. As you suggested, we plan to investigate the combined effect of UV-C containment unit during UV-C irradiation on other drug-resistant bacteria in addition to carbapenem-resistant E. cloacae and MRSA and conduct comparisons across bacterial species.

  1. Line 209-210 (clinical isolates of carbapenem-resistant E. cloacae and MRSA)>>>>>>> The author should describe more details of the source of these pathogens.

Response: We thank you for this comment. As you pointed out, we have included information on the carbapenem-resistant E. cloacae and MRSA strains used in this case as follows in the main text.

“Culture medium inoculated with the target bacterial species (clinical isolates of carbapenem-resistant E. cloacae and MRSA detected based on culture tests in patients who visited Tohoku University) was used as an environmental substitute carrier and placed in seven locations where UV-C was irradiated according to the specified method.” (Please see Page 7, Lines 223-226)

  1. Are all figures the standard model dimensions and location-wise?

Response: We thank you for this query. Regarding Figure 2, the UV-C irradiation device is displayed slightly larger to make it easier for readers to understand. The photograph in Figure 1 is provided to help readers better visualize the layout and dimensions of the room shown in Figures 2 and 3.

Round 2

Reviewer 2 Report

Comments and Suggestions for Authors

It can be accepted for the publication.

Reviewer 3 Report

Comments and Suggestions for Authors

Well done. The author responded to all raised questions.